# Mannose and glycine: Metabolites with potentially causal implications in chronic kidney disease pathogenesis

Yongzheng Hu, Wei Jiang *

Department of Nephrology, The Affiliated Hospital of Qingdao University, Qingdao, Shandong, China

* jiangwei866@qdu.edu.cn

## Abstract

### Background

Chronic Kidney Disease (CKD) represents a global health challenge, with its etiology and underlying mechanisms yet to be fully elucidated. Integrating genomics with metabolomics can offer insights into the putatively causal relationships between serum metabolites and CKD.

### Methods

Utilizing bidirectional Mendelian Randomization (MR), we assessed the putatively causal associations between 486 serum metabolites and CKD. Genetic data for these metabolites were sourced from comprehensive genome-wide association studies, and CKD data were obtained from the CKDGen Consortium.

### Results

Our analysis identified four metabolites with a robust association with CKD risk, of which mannose and glycine showed the most reliable causal relationships. Pathway analysis spotlighted five significant metabolic pathways, notably including "Methionine Metabolism" and "Arginine and Proline Metabolism", as key contributors to CKD pathogenesis.

### Conclusion

This study underscores the potential of certain serum metabolites as biomarkers for CKD and illuminates pivotal metabolic pathways in CKD's pathogenesis. Our findings lay the groundwork for potential therapeutic interventions and warrant further research for validation.

**Data Availability Statement:** Publicly available datasets were analyzed in this study. This data can be found here: http://ckdgen.imbi.uni-freiburg.de/. All the data generated by the MR analysis is in the included Supplementary Material.

**Funding:** This project was sponsored by the National Natural Science Foundation of China (NSFC 81870494). We would like to clarify that while the funding supported the research, the funder had no role in the study design, data collection and analysis, decision to publish, or preparation of the manuscript. The research team independently conducted all aspects of the study, ensuring that the analysis and findings were solely the result of the researchers' efforts and methodologies.

**Competing interests:** The authors have declared that no competing interests exist.

## Introduction

Chronic kidney disease (CKD) represents a significant global health challenge and is a primary contributor to the overall renal disease burden [1]. The World Health Organization (WHO) categorizes CKD as a major source of worldwide disability, attributing this to its widespread occurrence, enduring nature, and associated comorbidities [2]. Addressing CKD necessitates effective prevention and therapeutic strategies, underpinned by a profound understanding of its biological mechanisms [3]. Although genetic determinants are recognized contributors to CKD's pathogenesis [4], the disease's multifaceted nature means many of its biological intricacies remain elusive. Despite advancements in genetics, especially through genome-wide association studies (GWAS), translating genetic insights into tangible biological mechanisms presents a substantial hurdle [5].

Recent strides in omics technologies, notably metabolomics, have enriched our understanding of disease etiologies. Metabolomics, by elucidating altered metabolic pathways and intermediate metabolites, offers unprecedented insights into disease mechanisms [6]. Groundbreaking GWAS studies focused on metabolites have pinpointed disease-relevant loci, proposing that metabolites, as functional intermediaries, elucidate genetic underpinnings of renal disorders [7].

Metabolites, either as end products or intermediates of metabolic processes, hold pivotal roles in human physiology. The recent establishment of the genotype-dependent metabolic phenotypes database, which catalogues genetically determined metabolites (GDMs), through a comprehensive GWAS employing non-targeted metabolomics, sheds light on the intricate interplay between human serum metabolites and genetic variants in renal disease pathogenesis. Yet, a comprehensive understanding of GDMs, particularly in the context of CKD, remains incomplete, highlighting the need for in-depth analyses elucidating the interplay between genetic variations, metabolites, and CKD mechanisms.

Mendelian randomization (MR) serves as a robust epidemiological methodology, utilizing genetic variants as instrumental variables (IV) to establish definitive putatively causal links between exposures and outcomes. Owing to the inherent stability of genotypes, established at conception, MR provides robust estimates, largely impervious to confounding variables and reverse causation [8]. Capitalizing on this strength, MR, over the past decade, has been instrumental in delineating putatively causal relationships between risk exposures and diseases using GWAS summary statistics. The recent expansion of the metabolic spectrum through GWAS has culminated in a comprehensive atlas of GDMs.

In this study, we postulate the potential of this GDM atlas in elucidating the putatively causal relationship between GDMs and CKD. We employ a bidirectional MR methodology to: (i) ascertain the putatively causal impact of human serum metabolites on CKD, and (ii) discern metabolic pathways that could illuminate CKD's mechanistic underpinnings.

## Materials and methods

### Data source of metabolites

Genetic association data pertaining to serum metabolites were retrieved from the genome-wide association study (GWAS) server dedicated to metabolomics, accessible at https://metabolomips.org/gwas/ [9]. Shin et al. presented a seminal investigation into the genetic determinants of human metabolism by conducting a non-targeted metabolomics GWAS, identifying genetic associations for 486 serum metabolites [10]. Specifically, 7,824 participants from two European cohorts—the KORA F4 study in Germany (1,768 participants) and the UK Twin Study (6,056 participants)—were enlisted. Both studies secured approval from

respective ethics committees, with participants providing informed consent. Metabolomic profiling of fasting serum employed non-targeted mass spectrometry, with metabolite identification and relative quantification executed via Metabolon, Inc. (https://www.metabolon.com/). Following stringent quality control, 486 metabolites (309 known and 177 unidentified) were analyzed. The identified metabolites spanned eight biochemical categories, as classified by the Kyoto Encyclopedia of Genes and Genomes (KEGG) database. Comprehensive genotyping details for both cohorts can be found in prior publications [9, 10]. The meta-analysis encompassed approximately 2.1 million single nucleotide polymorphisms (SNPs).

## GWAS of chronic kidney disease (CKD)

Our study sought to elucidate the potential causal linkage between metabolites and CKD, using data from the CKDGen Consortium (http://ckdgen.imbi.uni-freiburg.de/) [11]. The potential impact of these genetic variants on the risk of CKD was meticulously examined using data from the fourth round of analyses by the CKDGen Consortium. Wuttke et al.'s study included a comprehensive cohort of 41,395 CKD cases and 439,303 controls, exclusively of European descent. This analysis was distinguished by its stringent quality control protocols and thorough imputation techniques. The consortium's design, participant recruitment, and genotyping methodologies have been previously detailed [12]. Participants underwent serum creatinine and BUN concentration measurements. Creatinine measurements via the Jaffé assay before 2009 underwent a 0.95 calibration multiplication. Adult participants (>18 years) had their GFR estimated using the Chronic Kidney Disease Epidemiology Collaboration (CKD-EPI) equation, while the Schwartz formula was employed for those 18 or younger. eGFR was winsorized between 15 and 200 ml min$^{-1}$ per 1.73 m$^2$. CKD was defined as an eGFR below 60 ml min$^{-1}$ per 1.73 m$^2$. For studies that reported blood urea measurements, BUN was derived by multiplying blood urea by 2.8, with units in mg dl$^{-1}$. Within the studies conducted under CKDGen, age and sex were systematically included as covariates. Every participant provided explicit written consent, and ethical clearance was obtained from the respective local authorities.

## Selection of instrumental variables (IVs)

Instrumental variables (IVs) in Mendelian randomization (MR) analysis must adhere to three core assumptions: (1) IVs are associated with the exposure (in this case, metabolites); (2) IVs influence the outcome (here, CKD) only through their effect on the exposure; and (3) IVs are not associated with any confounders [13]. In our study, selecting instrumental variables (IVs) for the 486 metabolites required a nuanced approach due to the varying availability of genome-wide significant variants. We initially identified genetic variants associated with the metabolites using a threshold of $P < 1 \times 10^{-5}$. This threshold was chosen specifically because the number of variants associated with metabolites at a genome-wide significance level ($P < 5 \times 10^{-8}$) was comparatively low. This approach ensures a balance between including a sufficient number of IVs for a robust analysis and maintaining statistical rigor. In contrast, for CKD, where genome-wide significant variants are more abundantly available, we employed a more stringent threshold of $P < 5 \times 10^{-8}$ for the association analysis. This differential threshold strategy was implemented to optimize the validity and strength of the instrumental variables in the context of the available genetic data for both metabolites and CKD, thereby enhancing the overall robustness of our results. Additionally, to evaluate the strength of the selected IVs, we assessed the explained variance ($R^2$) and the F statistic, adhering to the commonly recommended standard of $F > 10$ in MR analyses. During the linkage disequilibrium (LD) clumping process, we utilized the clumping functionality of PLINK software (version

v1.90) to ensure the independence of our IVs [14]. We set the LD threshold at r^2 < 0.01 within a 500 kb genomic window. The 1000 Genomes Project was used as the reference dataset for LD clumping, a widely recognized resource that enhances the credibility of our analysis. To maintain analytical rigor, we harmonized the SNPs linked to both the exposure and the outcome, ensuring alignment with the same effect allele. SNPs characterized by palindromic sequences and moderate allele frequencies, or those displaying incompatible alleles, were consistently excluded.

## Mendelian randomization design

We conducted two distinct MR analyses utilizing GWAS summary statistics to probe the bidirectional relationship between metabolites and CKD. The forward MR analysis treated metabolites as exposures and CKD as the outcome, while the reverse MR analysis considered CKD as the exposure and metabolites as the outcome. Initial MR analyses were performed employing a two-sample inverse variance weighted (IVW) method, which involves meta-analyzing SNP-specific Wald ratios via a random-effects inverse variance approach. This method allows for the weighting of each ratio by its standard error, simultaneously accounting for potential heterogeneity in measures. Should a single SNP within the metabolites instrument disproportionately influence the overall results, IVW analyses were systematically repeated, excluding SNPs one at a time. When all three instrumental variable assumptions are fulfilled, the IVW method yields consistent estimates of the exposure's putatively causal effect. However, contradictions among the instrumental variable assumptions may lead to incorrect inferences. Consequently, sensitivity analyses were executed. Alternative MR methods, including MR-Egger, weighted median, and weighted mode, known for robustness against directional pleiotropy, were applied to generate comparative estimates. MR-Egger permits certain SNPs to influence the outcome without altering the exposure [15], providing a formal test for directional pleiotropy [16]. The weighted median MR assumes validity for at least 50% of the SNPs, and weighted-mode MR clusters SNPs, computing an estimate based on the most populous cluster. The Q-test for IVW and MR-Egger served to identify potential violations of assumptions through the heterogeneity in associations among individual IVs [17]. Additionally, we undertook individual SNP analyses and leave-one-out analyses to gauge the likelihood of associations driven by individual SNP factors. All generated data files are included as Supplementary Files. These comprehensive analyses were executed using the TwoSampleMR(version 0.5.7) in R(4.2.3), with forest plots generated via the Forestplot package(1.1.1), ensuring a meticulous and transparent presentation of our results.

## Metabolic pathway analysis

We employed the web-based Metaconfict 5.0 platform (https://www.metaboanalyst.ca/) for metabolic pathway analyses [18]. The functional enrichment and pathway analyses modules were harnessed to discern metabolite pathways pertinent to CKD's biological processes. The Kyoto Encyclopedia of Genes and Genomes (KEGG) database was the primary resource, with a significance level set at 0.10 for pathway analysis. Only metabolites surpassing the recommended IVW association threshold (P_IVW_fixed & random<0.05) were considered.

## Ethics approval and consent to participate

This article contains human participants collected from several studies performed by previous studies. All participants gave informed consent in all the corresponding original studies. Our study is based on large-scale GWAS datasets, and not individual-level data. This study has

been registered and approved by the ethics review board of ethics committee approval of the affiliated hospital of Qingdao university. Ethics approval number: QYFY WZLL 28269.

## Results

### Instrumental variables (IVs) selection

For the set of 486 metabolites, the number of selected IVs varied, ranging from 3 to 479, with a median count of 15. Notably, the F statistics, which test the validity of IVs, all surpassed the threshold of 10. The lowest observed F statistic value was 17.411, signifying that the potential for weak instrumental bias is minimal. The full results of IVs selection were given in Additional file. The overarching design of this MR study is depicted in Fig 1.

### Causal effects of metabolites on CKD

In our study, we first applied the Inverse Variance Weighted (IVW) method to examine potential causal links between 486 metabolites, categorized into 9 distinct groups, and CKD. This initial phase, utilizing a p-value threshold of $<0.05$, revealed 46 metabolites potentially associated with CKD, comprising 14 unidentified and 32 identified compounds, as shown in S1 Fig (S1 Fig). Subsequently, to narrow down the metabolites that might have a positive causal influence on CKD, we incorporated an additional criterion: a positive beta coefficient (b_IVW > 0) in the IVW analysis. This further filtration decreased the number of relevant metabolites from 46 to 29. To enhance the reliability of our results and mitigate the risk of false positives from multiple testing, we subjected these 29 metabolites to rigorous evaluation using various MR techniques. These included the Weighted Median, Weighted Mode, and MR-Egger methods, while still adhering to a p-value cut-off of $<0.05$. After applying the false discovery rate (FDR) correction to adjust for multiple comparisons, our analysis continues to moderately support our initial findings, suggesting possible causal links between specific metabolites and CKD.

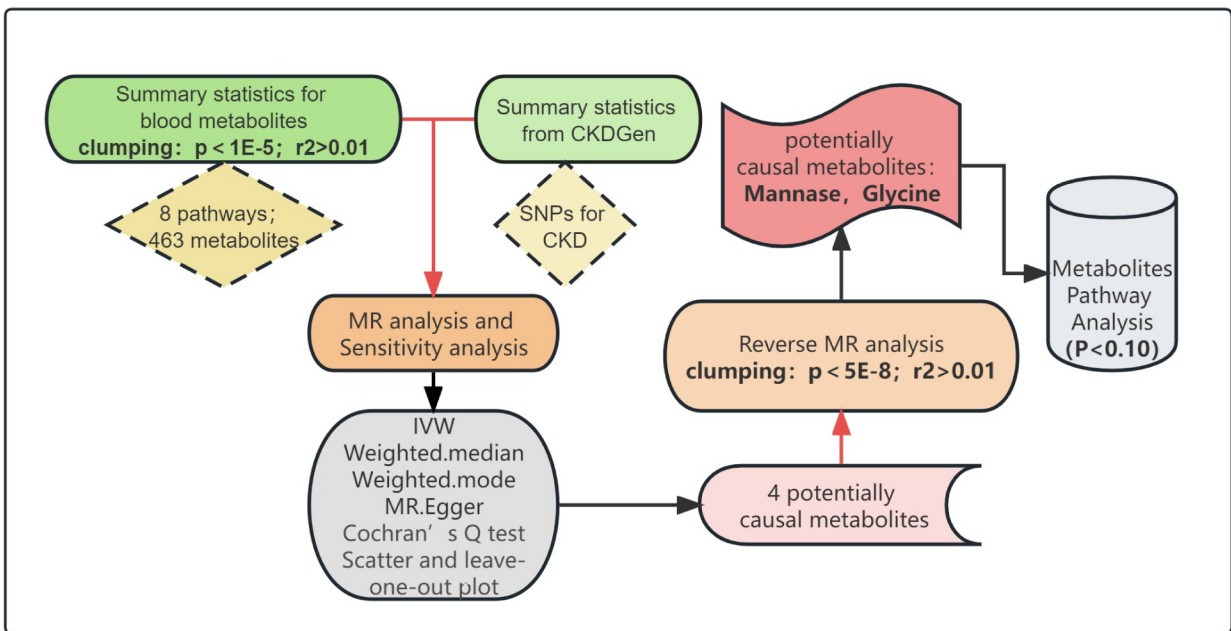

**Fig 1. The overview of the research workflow.** CKD: chronic kidney disease; SNP: single nucleotide polymorphism; MR: mendelian randomization; IVW: inverse variance weighted.

This comprehensive approach allowed us to discern 7 metabolites with strongly supported, plausible causal relationships with CKD. These metabolites were distributed across various pathways: Amino Acid Pathways: Proline: OR = 1.40, 95% CI: 1.12–1.74, p = 2.752E-03; N-acetylornithine: OR = 1.26, 95% CI: 1.14–1.40, p = 1.151E-05; Glycine: OR = 1.57, 95% CI: 1.25–1.98, p = 9.594E-05; Carbohydrate Pathways: Mannose: OR = 1.47, 95% CI: 1.07–2.02, p = 1.851E-02; Energy Pathways: Succinylcarnitine: OR = 1.35, 95% CI: 1.07–1.71, p = 1.241E-02; Unknown Pathways: X-12510: OR = 1.40, 95% CI: 1.13–1.74, p = 2.262E-03; X-12798: OR = 1.24, 95% CI: 1.02–1.51, p = 2.822E-02. The full results of suggestive associations are given in S2 Fig.

## Sensitivity analysis

To mitigate the potential influence of horizontal pleiotropy on MR estimates, sensitivity analyses were conducted. Fig 2 encapsulates the sensitivity results for the seven metabolite-CKD pairs with significant putatively causal associations. Generally, the putatively causal associations were deemed robust if they achieved statistical significance (p<0.05) in at least two additional MR tests, typically the Weighted Median and the Weighted.Mode tests. The MR-Egger's intercept term was utilized to identify potential horizontal pleiotropy across all associations. Notably, the associations of proline and X-12510 with CKD were significant in the Egger Intercept test, registering p-values of 4.572E-02 and 4.936E-02 respectively. Scatter (S3 Fig) and leave-one-out plots (S4 Fig) further confirmed the absence of potential outliers for all identified metabolites. We excluded the association between metabolite X-12798 and CKD due to the indication of heterogeneity via Cochran's Q test (Q_p_val = 1.079E-02). Consequently, four out of the seven pairs—mannose, N-acetylornithine, glycine, and succinylcarnitine with CKD—were identified as robust. A comprehensive report of the sensitivity and pleiotropy analysis is detailed in the supplementary files.

## Causal effects of CKD on metabolites

For CKD, 12 SNPs were chosen as instrumental variables (IVs). Upon application of the Cochran's Q test, ensuring the absence of heterogeneity (Q_p_val>0.05), the IVW-FE analysis revealed a non-significant putatively causal effect of CKD on the metabolite mannose, with an OR of 1.00 (95% CI: 0.98–1.02) and p-value of 8.878E-01. Comparable non-significant results were obtained for CKD's effect on glycine, with an OR of 0.99 (95% CI: 0.97–1.01) and p-value of 4.801E-01, even when using MR mixture models. In contrast, significant putatively causal effects were observed for CKD on N-acetylornithine (OR = 1.05, 95% CI: 1.01–1.09, p = 8.724E-03) and succinylcarnitine (OR = 1.02, 95% CI: 1.00–1.04, p = 3.643E-02) using IVW-FE, although these results were not consistently mirrored in other MR models(Fig 3). S5 and S6 Figs present the scatter plots with the regression line and funnel plots.

## Metabolic pathway analysis

We initially selected 32 metabolites for this analysis, using a p-value threshold of <0.05 according to the Inverse Variance Weighted (IVW) method. The comprehensive list of these metabolites can be found in the S6 Table. Subsequently, we used the Metaconflct 5.0 platform to analyze the selected 32 metabolites for their involvement in metabolic pathways. Upon excluding 14 that were unidentified and 4 that lacked a match, 14 metabolites were effectively recognized and identified by the platform. We discerned that the metabolites associated with CKD were predominantly concentrated in five pathways, all of which achieved significance (p < 0.10). These pathways are: "Methionine Metabolism" (p = 4.86E-03); "Arginine and Proline Metabolism" (p = 8.82E-03); "Betaine Metabolism" (p = 1.32E-02); "Fatty Acid

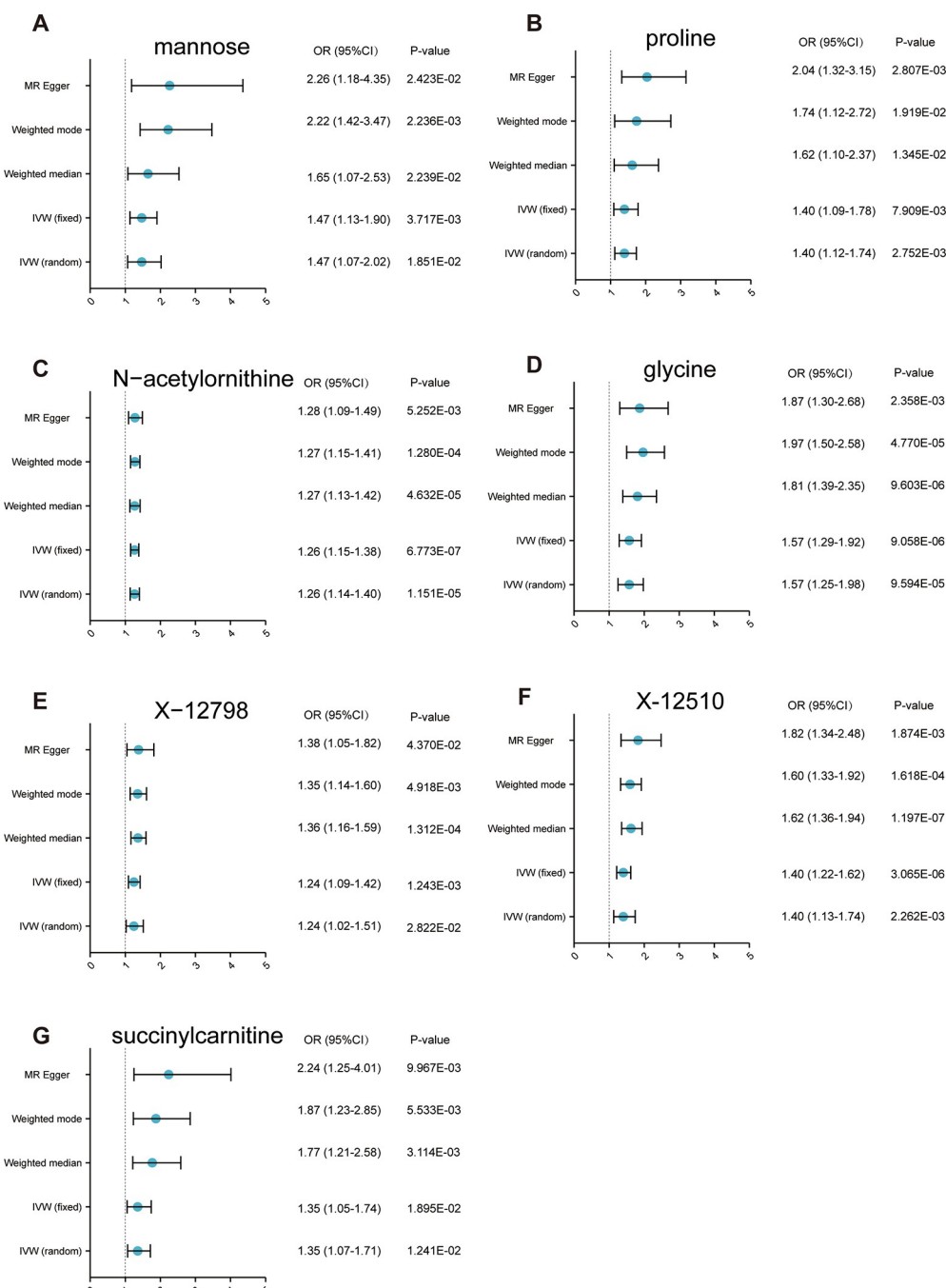

**Fig 2. Sensitivity analysis for significant metabolites on CKD.** OR: odds ratio; CKD: chronic kidney disease.

Biosynthesis" (p = 3.52E-02); "Glycine and Serine Metabolism" (p = 9.05E-02). The intricate interplay between these identified signaling pathways is visually represented in S7 Fig.

## Discussion

In this bidirectional Mendelian randomization study, we employed a rigorous, unbiased approach to investigate the putatively causal relationship between metabolites and CKD,

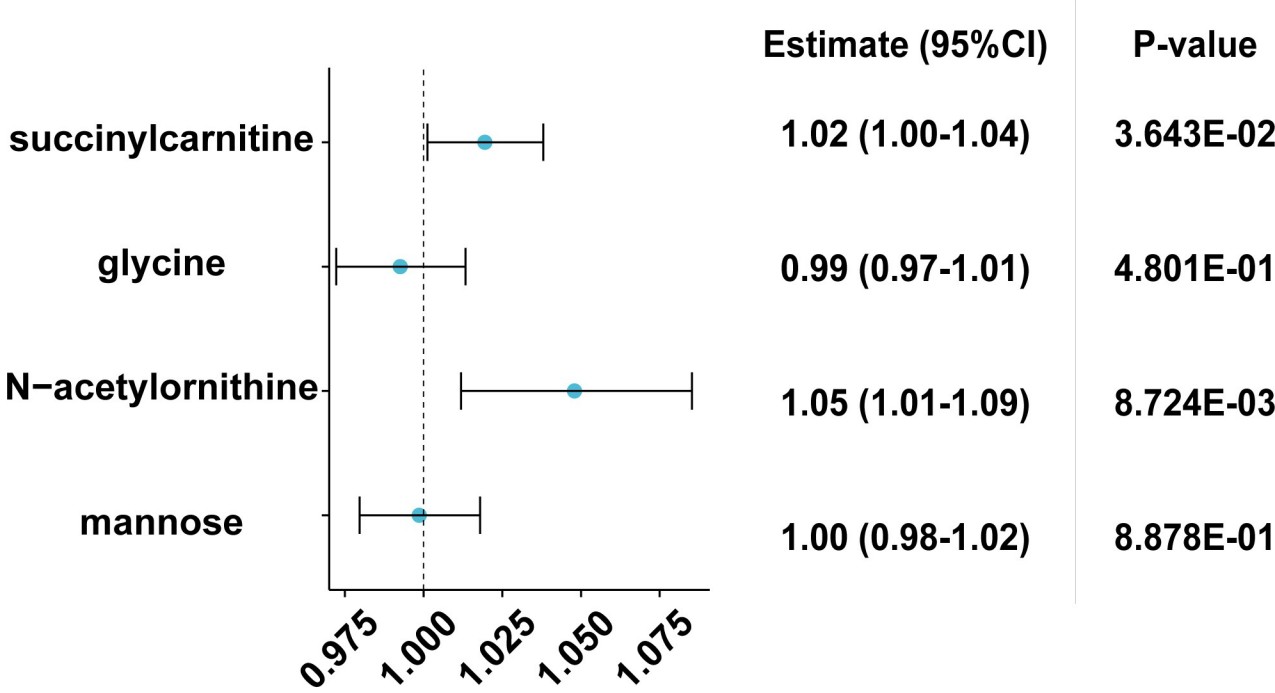

**Fig 3. Sensitivity analysis for CKD on the four metabolites selected.** CKD: chronic kidney disease.

leveraging the GWAS dataset from CKDGen. Out of 486 metabolites examined, seven were found to be relevant to CKD risk when probed using genetic variants. Notably, after a comprehensive sensitivity analysis, four of these metabolites emerged as particularly significant. Reverse Mendelian analysis further validated the robust putatively causal relationship of two metabolites, mannose and glycine, with CKD. Additionally, our pathway enrichment analysis highlighted five significant metabolic pathways: "Methionine Metabolism," "Arginine and Proline Metabolism," "Betaine Metabolism," "Fatty Acid Biosynthesis," and "Glycine and Serine Metabolism"—indicating their primary involvement in CKD's pathogenesis.

To the best of our knowledge, this is the inaugural bidirectional MR study that synergizes genomics with metabolomics to discern causality in CKD. Our findings spotlight a suite of serum metabolites that exhibit a relationship with CKD, with mannose and glycine standing out due to their pronounced impact on the disease.

D-Mannose, an isomer of glucose, is not recognized as an essential nutrient. Previous evidence from prospective cohorts indicates a significant correlation between elevated mannose levels and high risk of kidney failure requiring replacement therapy [19]. Concurrently, mannose has been employed as a biomarker to elucidate the observed association between the consumption of ultra-processed food and chronic kidney disease (CKD) [20]. These findings align with our own. However, the putatively causal relationship between mannose and CKD could be multifaceted. Certain research implicates mannose as a key factor driving γδ T cell infiltration in the kidney [21]. Kidney surface-associated mannose structures have been implicated in triggering γδ T cell recognition, instigating a pathogenic interleukin-17a (IL-17a)-mediated autoimmune response [22]. IL-17A, for instance, is implicated in the immunopathogenesis of crescentic glomerulonephritis (GN) by promoting neutrophil recruitment [23], and it plays a pivotal role in MPO-ANCA GN pathogenesis by fostering the development of MPO-specific αβ T cells [24]. Moreover, reports suggest that mannose, through induction of energy

expenditure via cellular ATP depletion, selectively modifies the synthesis of rapidly turning over glycosylated proteins such as PGs. This effect results in kidney size diminution, ureteric bud branching disorder, and subsequent kidney malformation [25]. Mannose is the primary monosaccharide involved in N-linked glycosylation, a form of post-translational modification. Exposure of mannose-glycosylated polysaccharides in lupus nephritis (LN) takes place on the surface of renal cells, thus enhancing the recognition by polysaccharide recognition receptors expressed on immune cells. Consequently, mannose glycosylation levels detected in renal biopsy specimens from LN patients have been validated as predictive markers for CKD progression [26]. Despite mannose's potential as a promising biomarker for CKD, further investigations are warranted to elucidate the underlying mechanisms.

Glycine, ubiquitous in all living organisms, can be synthesized in humans from threonine, choline, or hydroxyproline via organ interconversion metabolism primarily in the liver and kidneys. Despite its simple structure, glycine plays intricate roles within the human body. Interestingly, administration of glycine to rats demonstrated no observable effect on either the rate of glomerular filtration or renal blood flow [27]. Conversely, subsequent investigations suggested that glycine significantly augments renal hemodynamics through as-yet-undefined mechanisms, with secondary increases in excretion [28]. While there is evidence suggesting altered glycine metabolism in CKD disease models [29], the positive or negative impact of glycine on the kidney remains under debate. Historical proposals suggested a potential nephrotoxic effect of glycine [30], and recent findings indicate that glycine exacerbates ischemia-reperfusion-induced kidney injury via NMDA receptor agonism [31]. Notably, one of the facets of ischemia-reperfusion injury is the onset of oxidative stress. Yet, certain studies suggest glycine might inhibit oxidative stress onset by promoting the nuclear translocation of Nrf1, thus enhancing Glo2 function [32], which could mitigate kidney injury in models induced by lead [33] or cobalt chloride [34]. Furthermore, CKD development has been linked to ferroptosis [35], a process that glycine can potentiate via SAM-mediated GPX4 promoter methylation [36]. Intriguingly, glycine also contributes to the synthesis of glutathione (GSH), an antioxidant implicated in ferroptosis inhibition [37]. Moreover, glycine demonstrates a robust association with CKD, presenting potential predictive value for significant health outcomes that may impact CKD.

It's worth noting that the kidneys contribute to glucose synthesis from amino acids via gluconeogenesis, regulate acid-base balance through inter-organ metabolism of glutamine, and synthesize amino acids like arginine, tyrosine, and glycine. Renal dysfunction can lead to an imbalance in these metabolic processes, promoting muscle protein breakdown, inflammation, mitochondrial abnormalities, immune response defects, and cardiovascular diseases [38]. Another study elaborates on how the progressive loss of kidney function in CKD patients is associated with several complications, including metabolic acidosis, malnutrition, and protein-energy wasting. Altered kidney amino acid and protein metabolism could play a key role in the homeostasis of various vasoactive compounds and hormones in patients with advanced disease [39]. Based on these insights, it's evident that the findings of our study, which identified a strong association between certain amino acids (like proline and glycine) and CKD, might be reflective of the kidney's diminished capacity to metabolize and excrete these amino acids. This metabolic impairment, rather than the amino acids themselves, could contribute to their increased serum levels in CKD.

In this study, the metabolic pathway analysis showed that "Methionine Metabolism", "Arginine and Proline Metabolism", "Betaine Metabolism", "Fatty Acid Biosynthesis" and "Glycine and Serine Metabolism" pathways are mainly associated with CKD.

Notably, Methionine, an aliphatic and sulfur-containing essential amino acid, has been implicated in lipid metabolism, the activation of endogenous antioxidant enzymes such as

methionine sulfoxide reductase A, and glutathione biosynthesis [40]. An escalation in ROS levels in the kidneys was noted in the MetO group and the Met+MetO group, denoting heightened oxidative stress levels [41]. Other research has noted an increase in tHcy and a concomitant decrease in Met levels in the blood as CKD progresses. In particular, a decline in GFR, especially in the presence of proteinuria, is associated with reduced Met levels [42]. The upregulation of NNMT expression in the kidneys has been linked to disturbances in methionine metabolism and consequent renal fibrosis [43]. Moreover, methionine serves as a precursor of homocysteine and cystathionine. Hyperhomocysteinemia (HHcy) and raised homocystinuria levels are closely associated with chronic renal failure [44]. Taken together, these findings strongly suggest a critical role for methionine metabolism in the biological mechanisms underlying CKD.

Evidence has underscored the role of proximal convoluted tubules as the primary site for endogenous arginine biosynthesis [45]. Studies have highlighted variations in arginine and proline metabolism across a range of contexts, such as a Garidi-5-induced rat model of subacute hepato-renal toxicity [46], a chromium exposure-induced injury model [47], an adenine-induced chronic kidney disease model [48], and in instances of diabetic nephropathy [49].

Betaine, a critical osmolyte, accumulates in the basolateral plasma membrane of medullary epithelial cells via the betaine/gamma-aminobutyric acid transporter (BGT1) [50]. Its synthesis is augmented under hyperosmotic conditions [51]. Notably, serum osmolality has emerged as an independent risk factor for CKD onset [52]. In a retrospective longitudinal investigation, a higher baseline urinary osmolality was found to be closely linked to an elevated risk of end-stage kidney disease [53].

Ectopic lipid accumulation within the kidneys has emerged as a characteristic feature of metabolic conditions precipitating chronic kidney disease (CKD) [54]. Persistent disturbances in fatty acid metabolism have been observed in murine CKD models. A heightened fatty acid biosynthesis, coupled with downregulated triglyceride and diglyceride catabolic pathways, correlates with a notable increase in serum lipid levels [55]. Accumulation of fatty acids in the kidneys is purportedly deleterious to podocytes, precipitating cell apoptosis and subsequent glomerulosclerosis [56]. Lipidomic investigations have illuminated the pivotal role of the structural allocation of free fatty acids between fatty acid oxidation and complex lipid fatty acid composition in driving CKD progression [57].

In response to reduced urinary excretion due to diminished GFR, plasma levels of D-serine increase. Consequently, plasma D-serine levels are integral to kidney disease detection, with its concentration potentially escalating alongside CKD progression [58]. Notably, a combination of plasma and urinary D-serine levels serves to differentiate CKD origins, particularly pertinent in the context of lupus nephritis [59]. Collectively, these findings suggest a potential relevance of these metabolites' biosynthesis and metabolism to the biological mechanisms underpinning CKD.

However, this study faces several limitations: First, the observed associations in our study may not necessarily indicate a direct causative relationship between these amino acids and CKD. Instead, they could represent a secondary effect of CKD-related metabolic dysregulation. This perspective is crucial for understanding the complex interplay between kidney function, amino acid metabolism, and CKD progression. Future research should aim to unravel these intricate relationships and explore potential therapeutic interventions that target metabolic disturbances in CKD. Second, while our analyses revealed potential causal relationships between certain metabolites and CKD, future studies should include more stringent statistical correction methods, such as the False Discovery Rate (FDR) adjustment, to properly address the issue of multiple testing. Such correction is not only crucial for reducing false positives but also enhances our understanding of which findings are truly biologically significant. Third,

The efficacy of the instrumental variables (IVs) is closely tied to the GWASs' sample size. A larger dataset would bolster accuracy. While Mendelian randomization is a potent tool for discerning causality between human blood metabolites and CKD, experimental data-based studies are essential for validation. The reliability of our MR analysis is largely contingent on the instrumental variables' explanatory power on exposure, emphasizing the need for a more expansive sample size. Last, for pathway analysis, due to data constraints, we included metabolites with P values > 0.10. While we identified CKD-risk-contributing metabolites, their roles in CKD's pathogenic mechanism warrant further exploration.

## Conclusions

In our comprehensive bidirectional MR analysis, we identified mannose and glycine as metabolites with a strong putatively causal association with CKD. We further spotlighted five metabolic pathways that may underpin CKD's etiology. These insights pave the way for harnessing select metabolites as diagnostic biomarkers and for devising targeted treatment strategies. Nevertheless, further research is paramount to corroborate our findings.

## Supporting information

**S1 Fig. Comprehensive assessment of metabolite associations with CKD.** (A) Bar chart displaying the distribution of metabolites, categorized and color-coded by their respective classes. (B) A volcano plot visualizing the relationship between metabolites and CKD as assessed by the IVW algorithm. Individual dots signify specific metabolites, with the dot size reflecting the count of associated SNPs for each metabolite. Red dots indicate a positive association with CKD, while blue signifies a negative association. The red dashed line demarcates the statistical significance threshold (P = 0.05). IVW: inverse variance weighted; SNP: single nucleotide polymorphism; CKD: chronic kidney disease.
(TIF)

**S2 Fig. The analysis results of all metabolites with p<0.05 obtained by the IVW algorithm.** IVW: inverse variance weighted; SNP: single nucleotide polymorphism.
(TIF)

**S3 Fig. The genetic associations of seven metabolites on the risk of CKD.** IVW: inverse variance weighted; CKD: chronic kidney disease.
(TIF)

**S4 Fig. Leave-one-out plots for 7 metabolites.**
(TIF)

**S5 Fig. The genetic associations of CKD on selected metabolites.** IVW: inverse variance weighted; CKD: chronic kidney disease.
(TIF)

**S6 Fig. The funnel plot represents IVs for each significant potentially causal association between CKD and selected metabolites.** CKD: chronic kidney disease.
(TIF)

**S7 Fig. Metabolism pathway analysis.** (A) Dot plots show the top 25 metabolic pathways in which significant metabolites participate, selected by the IVW algorithm. (B) Network diagram, individual nodes denote distinct metabolite sets. An edge or a connection between two metabolite sets indicates that they have an overlap, wherein more than 25% of their metabolites

are shared.
(TIF)

**S1 Table. Metabolites names list.**
(CSV)

**S2 Table. Forward MR analyses of potentially causal associations between metabolites SNP and CKD.**
(XLS)

**S3 Table. Exposure-CKD-clumped selected SNP.**
(XLSX)

**S4 Table. Reverse MR analyses of potentially causal associations between each CKD SNP and metabolites.**
(XLS)

**S5 Table. Metabolic pathways analysis results.**
(XLSX)

**S6 Table. Pathways involved metabolites.**
(XLSX)

**S7 Table. The false discovery rate results in IVW methods.**
(CSV)

## Acknowledgments

We thank all the consortium studies for making the summary association statistics data publicly available.We also thank Yi Yu for his help in coding analysis.

## Author Contributions

**Conceptualization:** Yongzheng Hu, Wei Jiang.

**Data curation:** Wei Jiang.

**Formal analysis:** Yongzheng Hu.

**Investigation:** Wei Jiang.

**Methodology:** Yongzheng Hu.

**Project administration:** Wei Jiang.

**Resources:** Yongzheng Hu.

**Software:** Yongzheng Hu.

**Supervision:** Wei Jiang.

**Validation:** Yongzheng Hu, Wei Jiang.

**Visualization:** Yongzheng Hu.

**Writing – original draft:** Yongzheng Hu.

**Writing – review & editing:** Wei Jiang.

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
