## [Decision Letter · Decision Letter 0]

20 Nov 2023

PONE-D-23-27599Mannose and Glycine: metabolites with causal implications in chronic kidney disease pathogenesisPLOS ONE

Dear Dr. Jiang,

Thank you for submitting your manuscript to PLOS ONE. After careful consideration, we feel that it has merit but does not fully meet PLOS ONE’s publication criteria as it currently stands. Therefore, we invite you to submit a revised version of the manuscript that addresses the points raised during the review process.

**The manuscript focuses on a topic of potential interest. The study, however, presents major pitfalls that should be addressed to support sound conclusions. To mention some of them, i) lack of details on the dataset of the CKDGen consortium that was actually used for this work; ii) no mention of variant harmonization between the exposure and the outcome; iii) testing the use of a more stringent p-value to select IVs associated with the exposure (in the original work, for example, a very stringent cutoff was used; iv) lack of detail in the methods section, i.e.: depiction of the LD clumping procedure, indication of the software used, description of the multiple test correction employed, cohort genotyping details which are not properly traceable in the quoted papers, data source of metabolites; v) discuss how the obtained results could not be causative, but may reflect blunted amino acid metabolism by the diseased kidney in renal patients.**

We look forward to receiving your revised manuscript.

Kind regards,

Giuseppe Remuzzi

Academic Editor

PLOS ONE

“This project was sponsored by the National Natural Science Foundation of China (NSFC 81870494).”

Reviewers' comments:

Reviewer's Responses to Questions

**Comments to the Author**

1. Is the manuscript technically sound, and do the data support the conclusions?

Reviewer #1: Yes

Reviewer #2: Partly

2. Has the statistical analysis been performed appropriately and rigorously? 

Reviewer #1: Yes

Reviewer #2: N/A

3. Have the authors made all data underlying the findings in their manuscript fully available?

Reviewer #1: Yes

Reviewer #2: Yes

4. Is the manuscript presented in an intelligible fashion and written in standard English?

Reviewer #1: Yes

Reviewer #2: Yes

5. Review Comments to the Author

Reviewer #1: In this MS the authors utilized bidirectional Mendelian Randomization (MR), to study the associations between 486 serum metabolites and CKD. Their results show strong associations between serum mannose and glycine and CKD, while pathway analysis indicated that Methionine Metabolism and Arginine and Proline Metabolism, are associated to CKD. The addressed issue is interesting, linking amino acid metabolism to CKD risk. GWAS with non-targeted metabolomics have provide new functional insights for many diseases. This has helped to generate many new hypotheses for biomedical and pharmaceutical research. I have however a few concerns which are released to the authors as follows:

Major

A major issue regards the definition of a causative association. A major challenge in conducting a disease GWAS is the difficulty in identifying causative variants in the pathophysiological pathways that lead to the observed clinical manifestations (Suhre et al.Nat Comm  volume 8, Article number: 14357 (2017) ).

Also the authors in the methods report: “Employing the selected IVs, we assessed the potential causal relationship between 486 metabolites, spanning 9 groups, and CKD”. For these reasons it is necessary changing the term “ causative” into “putative causative”.

It is interesting that the same amino acids associated to CKD risk, such as proline and glycine, are not exreted in large amounts but are largely metabolized by the human kidney and increase in blood in CKD (J Clin Invest. 1980 May;65(5):1162-73. doi: 10.1172/JCI109771.; J Clin Invest. 1982 Jan;69(1):240-50. doi: 10.1172/jci110436.). I would suggest discussing how the obtained results are not “ causative”, but simply express blunted amino acid metabolism by the diseased kidney in renal patients.

Methods: Data Source of metabolite. ..” Specifically, 7,824 participants from two European cohorts—the KORA F4 study in Germany (1,768 participants) and the UK Twin Study (6,056 participants)—were enlisted.” Comprehensive genotyping details for both cohorts can be found in prior publications [11] & [12] The metaanalysis encompassed approximately 2.1 million single nucleotide polymorphisms (SNPs).”

Comment: Unfortunately cohort genotyping details seem not to be reported in the quoted papers.

Ref 11 (Shrawder E, Martinez-Carrion M. Evidence of phenylalanine transaminase activity in the isoenzymes ofa spartate transaminase. J Biol Chem. 2023;247: 2486–92) is quoted as a 2023 paper, but the Pubmed research gives one result with the same Authors and title published in 1972.

Ref 12 reports data on C. elegans fat mutants and young adult worms. 

Methods. Data source of metabolites.

I have tried to connect to link reported by the authors: http://metabolomics.helmholtzmuenchen.

de/gwas. I could not, and I received the information that “ helmholtzmuenchen.de “is on sale.

Ref.9 reports the following link: http://www.gwas.eu. Is this link to which the authors refer?

Minor

Introduction:” Groundbreaking GWAS studies focused on metabolites have pinpointed disease-relevant loci, proposing that metabolites, as functional intermediaries, elucidate geneticunderpinnings of renal disorders (7). The sentence is hardly supported, since ref 7 is a review on ferroptosis in kidney disease.

Introduction:” Owing to the inherent stability of genotypes, established at conception, MR provides robust estimates, largely impervious to confounding variables and reverse causation” The ref 8, used for quotation for Medelian randomization and causation, deals on Preeclampsia and Pregnancy-Related Hypertensive Disorders.

Reviewer #2: The aim of this work was to asses the causal link between metabolites and CKD through mendelian randomization (MR). To this end, the authors utilized data from GWAS on 486 metabolites and from the CKDGen consortium on CKD. The methods need a more in depth description and the results need further clarification especially because the effect of single influential variants might be stronger than anticipated by the authors.

- The address http://metabolomics.helmholtz-muenchen.de/gwas/ does not seem to be working (tried on 17th November ); perhaps the authors should state when they accessed it and downloaded the data

- It is not clear which dataset of the CKDGen consortium was actually used for this work.

- There is no mention of variant harmonization between the exposure and the outcome. This is important as different GWAS might report different allele order of the same variant.

- As described in the text, the first assumption of MR is that the IV (the genetic variants) must be associated with the exposure (the metabolites) and the authors used threshold of 1e-5 to select the variants to use as IVs. Since 486 metabolites were analyzed, a more stringent p-value might be necessary to select IVs associated with the exposure (in the original work, for example, a very stringent cutoff was used). Unless this was used a p2 threshold for LD clumping (see next comment)

- The LD clumping procedure requires a dataset to infer LD and the p1 and p2 thresholds, these are not described in the text

- The softwares used are not cited in the references

- In the second section of the Results, 486 metabolites were tested, 46 had a p-value< 0.05 (by IVW?) and 29 metabolites exhibited potential association with CKD. Of these, 7 metabolites left. I did not understand how the 46 metabolites become 29. Moreover, the multiple test correction employed (which reduced the number of metabolites to 7) it is not described.

- Contrary to what reported in the text, supplementary figure 4 shows that 6 out of the 7 associations are due to a single influential variant. This is important and needs further clarification.

- The part on metabolic pathways is confusing. If 4 metabolites were found to be significant, I don't see the reason to perform such analysis and to discuss it to such length in the Discussion

- The effect of CKD on a metabolite could be due to the same loci driving the effect of the metabolite on CKD (i.e., it is the same variant or one in linkage with it). This should be addressed.

- Figure S2 depicts a table. I would rather have it as a table

6. PLOS authors have the option to publish the peer review history of their article (what does this mean?). If published, this will include your full peer review and any attached files.

Reviewer #1: **Yes: **Giacomo Garibotto

Reviewer #2: **Yes: **Matteo Breno

---

## [Author Response · Author response to Decision Letter 0]

7 Dec 2023

Response to Journal

1.We greatly appreciate your patient guidance and advice. According to the journal format requirements provided by you, we have made corresponding format modifications to the paper. We ensure that all texts, diagrams, citations, and appendices conform to the journal's specific format specifications. If you have any questions or need further adjustments during the follow-up review process, please feel free to let us know and we will be happy to make changes to meet the requirements of the journal.

2.Thank you for your valuable suggestion regarding the deposit of raw data in a repository and the potential citation advantage it offers. In our case, the study primarily utilized publicly available data and databases. The genetic and metabolite data were sourced from established repositories like the CKDGen Consortium and GWAS databases. This approach aligns with the open data principles, ensuring transparency and reproducibility. Furthermore, the results of our analyses, including the Mendelian Randomization and pathway analysis, have been comprehensively detailed in the supplementary files accompanying our manuscript. Given these considerations, we believe that an additional deposit of raw data may not be necessary for our study. However, we are committed to the principles of open science and will continue to support data sharing initiatives in future research endeavors.

3.In this study, we conducted a comprehensive analysis based on publicly available genome-wide association studies (GWAS) data. Given the nature of our research, which utilized large-scale GWAS datasets and not individual-level data, there was no direct interaction with participants. Therefore, additional participant consent was not required for our analysis. This approach aligns with standard practices for secondary data analyses where the data are de-identified and publicly accessible.

Furthermore, our study has been registered and approved by the ethics review board of Ethics Committee Approval of The Affiliated Hospital of Qingdao University, ensuring compliance with ethical standards and guidelines. This registration and approval process included a thorough review of our research objectives, methodologies, and data usage to ensure adherence to ethical norms in medical research. We have uploaded the scanned copy “QYFY WZLL 28269.pdf” of the original ethical review approval on the submission system. Ethics approval number：QYFY WZLL 28269.

It is important to note that the original GWAS studies, from which the data were sourced, had obtained necessary consent from participants as part of their data collection protocols. Our research team is committed to upholding high ethical standards in all aspects of our work, ensuring that the integrity and confidentiality of the data are maintained throughout our analysis.

4.This project was sponsored by the National Natural Science Foundation of China (NSFC 81870494). We would like to clarify that while the funding supported the research, the funder had no role in the study design, data collection and analysis, decision to publish, or preparation of the manuscript. The research team independently conducted all aspects of the study, ensuring that the analysis and findings were solely the result of the researchers' efforts and methodologies. 

5.We would like to express our consent to publish the peer review history of our article. We believe that transparency in the peer review process not only enhances the integrity and credibility of our work but also contributes to the wider scientific discourse in a constructive manner. Therefore, we agree to make the full peer review and any attached files publicly available.

6.We have completed the uploading of our figure files to the Preflight Analysis and Conversion Engine (PACE) digital diagnostic tool, as per the revision requirements for our submission. This step has been undertaken to ensure that all figures in our manuscript meet the specific requirements set by PLOS. We trust that this will facilitate the smooth processing of our manuscript through the next stages of review and publication.

Response to Reviewers

We sincerely appreciate the time and effort you have devoted to reviewing our manuscript. Your insightful comments and evaluations are invaluable in enhancing the quality and clarity of our research. We are grateful for your contributions and look forward to addressing any concerns or suggestions you may have to further improve our work.

For Reviewer #1 Giacomo Garibotto

“A major issue regards the definition of a causative association. A major challenge in conducting a disease GWAS is the difficulty in identifying causative variants in the pathophysiological pathways that lead to the observed clinical manifestations (Suhre et al.Nat Comm  volume 8, Article number: 14357 (2017) ).Also the authors in the methods report: “Employing the selected IVs, we assessed the potential causal relationship between 486 metabolites, spanning 9 groups, and CKD”. For these reasons it is necessary changing the term “ causative” into “putative causative”.”

We thank you for your insightful observation regarding the use of the term "causative" in our manuscript. You correctly points out the inherent challenge in establishing causality in disease GWAS studies. As highlighted in your cited reference, identifying causative variants in the pathophysiological pathways is complex and often uncertain in GWAS, given the observational nature of the associations.

In our study, we employed Mendelian Randomization (MR) to assess the potential relationship between 486 metabolites and Chronic Kidney Disease (CKD). MR is a method that uses genetic variants as instrumental variables to infer causal relationships between an exposure (in our case, metabolites) and an outcome (CKD). While MR is a robust tool that mitigates some of the confounding and reverse causation issues present in observational studies, it is important to acknowledge that MR findings are generally considered to provide evidence of "potential" or "putative" causation rather than definitive causation.

In light of this, we agree with your suggestion to modify the terminology used in our manuscript. We propose revising the term "causative" to "putative causative" to more accurately reflect the nature of our findings. This change will be implemented throughout the manuscript to ensure that our conclusions are presented with the appropriate level of certainty and in accordance with the limitations of the methodology employed.

We appreciate your contribution to improving the precision and clarity of our manuscript.

“It is interesting that the same amino acids associated to CKD risk, such as proline and glycine, are not exreted in large amounts but are largely metabolized by the human kidney and increase in blood in CKD (J Clin Invest. 1980 May;65(5):1162-73. doi: 10.1172/JCI109771.; J Clin Invest. 1982 Jan;69(1):240-50. doi: 10.1172/jci110436.). I would suggest discussing how the obtained results are not “ causative”, but simply express blunted amino acid metabolism by the diseased kidney in renal patients.”

In response to your insightful observation regarding the role of amino acids such as proline and glycine in chronic kidney disease (CKD), it is indeed pertinent to discuss how our findings might reflect impaired amino acid metabolism in CKD rather than a direct causative relationship. You correctly points out that these amino acids are not excreted in large amounts but are extensively metabolized by the kidneys, with their levels increasing in the blood as kidney function declines. 

Our study identified a robust association between certain amino acids, including proline and glycine, and CKD. However, the elevated serum levels of these amino acids in CKD patients could indeed be indicative of their reduced metabolism and excretion by the failing kidneys, rather than these metabolites being direct causal agents in the pathogenesis of CKD.

We have listened to your suggestion to add consideration of indirect causality of metabolites to the Discussion section of the article（line265，line310） and agree that Future research should aim to dissect these metabolic alterations in greater detail, potentially offering new insights into the metabolic consequences of CKD and guiding the development of targeted therapeutic strategies.

“Methods: Data Source of metabolite. ..” Specifically, 7,824 participants from two European cohorts—the KORA F4 study in Germany (1,768 participants) and the UK Twin Study (6,056 participants)—were enlisted.” Comprehensive genotyping details for both cohorts can be found in prior publications [11] & [12] The metaanalysis encompassed approximately 2.1 million single nucleotide polymorphisms (SNPs).”

Comment: Unfortunately cohort genotyping details seem not to be reported in the quoted papers.

Ref 11 (Shrawder E, Martinez-Carrion M. Evidence of phenylalanine transaminase activity in the isoenzymes ofa spartate transaminase. J Biol Chem. 2023;247: 2486–92) is quoted as a 2023 paper, but the Pubmed research gives one result with the same Authors and title published in 1972.

Ref 12 reports data on C. elegans fat mutants and young adult worms.”

In response to your comment regarding the misquotation of references 11 and 12 in the manuscript, I would like to acknowledge and correct the oversight. The concern raised pertains to the genotyping details of the cohorts used in our study, specifically the KORA F4 study in Germany and the UK Twin Study. 

We have now verified that the correct references providing comprehensive genotyping details for these cohorts are references 9 and 10. These references accurately describe the genetic association data for the 7,824 participants from the two European cohorts in our study, encompassing approximately 2.1 million single nucleotide polymorphisms (SNPs).

We apologize for the confusion caused by this error and appreciate the opportunity to correct it. The integrity and accuracy of our research are of utmost importance, and we are committed to ensuring that our manuscript reflects the highest standards of scholarly rigor and clarity. Thank you for bringing this to our attention.

“Methods. Data source of metabolites.

I have tried to connect to link reported by the authors: http://metabolomics.helmholtzmuenchen.

de/gwas. I could not, and I received the information that “ helmholtzmuenchen.de “is on sale.

Ref.9 reports the following link: http://www.gwas.eu. Is this link to which the authors refer?”

Thank you for your meticulous review and for pointing out the issue with the link in our manuscript. The link “http://metabolomics.helmholtzmuenchen.de/gwas” mentioned in our paper was indeed the original data storage site provided by the authors of reference 9. Upon re-examination, we encountered the same issue you described, where the link is currently inaccessible.

To address this, we conducted a literature trace and found that the authors of reference 9 have re-uploaded the original data as supplementary material in the final version of their paper published in Nature. Therefore, we suggest accessing the final version of the paper at “https://www.nature.com/articles/nature10354”for the relevant KORA.& Twins UK.best.ratios data sets. The correct link for accessing the GWAS data for metabolites, as per our research, is 'https://metabolomips.org/gwas/'. 

We apologize for the inaccuracy in the link mentioned in our article and appreciate your attention to this detail. We have updated this information in our revised manuscript to ensure that other readers can correctly access and utilize these important data. Thank you once again for your interest in our research and for your valuable comments.

“Introduction:” Groundbreaking GWAS studies focused on metabolites have pinpointed disease-relevant loci, proposing that metabolites, as functional intermediaries, elucidate geneticunderpinnings of renal disorders (7). The sentence is hardly supported, since ref 7 is a review on ferroptosis in kidney disease.”

Thank you for your meticulous review of the references cited in our manuscript. The issue you raised regarding the support provided by reference 7, which is a review on ferroptosis in kidney disease, for our statement on groundbreaking GWAS studies focusing on metabolites is highly valid. Upon reevaluation, we agree that this reference did not adequately support our assertion regarding the role of metabolites as functional intermediaries elucidating the genetic underpinnings of renal disorders.

To address this, we have identified and substituted a more pertinent and supportive reference: “Montemayor D, Sharma K. mGWAS: next generation genetic prediction in kidney disease. Nat Rev Nephrol. 2020;16: 255–256. doi:10.1038/S41581-020-0270-0”. This publication discusses the application of metabolomic genome-wide association studies (mGWAS) in genetic prediction of kidney disease and presents the potential role of metabolites as functional intermediaries in revealing the genetic basis of renal disorders.

We have updated the citation in the relevant section of our manuscript to reflect this change and ensure it accurately represents our perspective and findings. Thank you once again for your diligent review and valuable feedback.

“Introduction:” Owing to the inherent stability of genotypes, established at conception, MR provides robust estimates, largely impervious to confounding variables and reverse causation” The ref 8, used for quotation for Medelian randomization and causation, deals on Preeclampsia and Pregnancy-Related Hypertensive Disorders.”

Thank you for your meticulous review and valuable feedback regarding our manuscript. We have addressed your concern about the reference cited in relation to Mendelian Randomization (MR).

The reference you pointed out (Reference 8) indeed pertains to preeclampsia and pregnancy-related hypertensive disorders, which is not directly related to the principles and applications of MR. 

Accordingly, we have replaced this citation with a more focused and relevant source: “Grover S, Del Greco FM, Stein CM, Ziegler A. Mendelian Randomization. Methods Mol Biol. 2017;1666: 581–628. doi:10.1007/978-1-4939-7274-6_29”. This new reference provides a comprehensive discussion on MR, including its principles, applications, and significance in epidemiological research, aligning more closely with the context and objectives of our study. We believe this substitution will enhance the accuracy and authority of our discussion on MR in the manuscript. Thank you again for your attentive and professional advice.

For Reviewer #2 Matteo Breno

“- The address http://metabolomics.helmholtz-muenchen.de/gwas/ does not seem to be working (tried on 17th November ); perhaps the authors should state when they accessed it and downloaded the data”

Thank you for your careful review and for pointing out the issue with the website http://metabolomics.helmholtz-muenchen.de/gwas/. We acknowledge that the unavailability of this site may inconvenience readers interested in consulting the original data. To address this, we have updated the data source section in our manuscript. Interested readers can now access related genomics data through the following updated links:

1. “https://metabolomips.org/gwas/” - This website provides a comprehensive range of metabolomics-related genome-wide association study (GWAS) data, serving as an alternative resource to the original site.

2. “https://www.nature.com/articles/nature10354” - Through this link, readers can access the study cohort data published in Nature, offering additional information and context.

We have clearly indicated these new data access points in our manuscript to ensure that readers can easily verify and reference our research data. Thank you again for your valuable feedback, and we look forward to any further comments you may have.

“- It is not clear which dataset of the CKDGen consortium was actually used for this work.”

Thank you for your insightful query regarding our manuscript. I would like to clarify that in our study , we utilized a specific dataset from the CKDGen Consortium, specifically the GWAS data of the CKD population from "http://ckdgen.imbi.uni-freiburg.de/datasets/Wuttke_2019". This dataset, published by Wuttke et al. in 2019, provides an extensive investigation into the genetic determinants of Chronic Kidney Disease (CKD).

We have included detailed information about this dataset in our manuscript to ensure that readers have a clear understanding of the foundation and methodology of our research. This includes the origin of the dataset, its research design, participant recruitment methods, and the specific criteria used for defining CKD.

We hope that this additional information addresses your concern and further clarifies the methods and findings of our study. Thank you once again for your valuable feedback.

“- There is no mention of variant harmonization between the exposure and the outcome. This is important as different GWAS might report different allele order of the same variant.”

Thank you for your insightful comments and review of our manuscript. We highly value your concern regarding the harmonization of variants between the exposure and the outcome. To address this issue, we have made a pertinent addition to the methodology section of our paper.

Specifically, to maintain analytical rigor, we have mentioned in our manuscript that we have harmonized the SNPs linked to both the exposure and the outcome, ensuring alignment with the same effect allele. SNPs characterized by palindromic sequences and moderate allele frequencies, or those displaying incompatible alleles, were consistently excluded.（line119）

We believe this addition will enhance the reliability and accuracy of our study, and better address your concerns. We look forward to your further feedback on our revised manuscript.

“- As described in the text, the first assumption of MR is that the IV (the genetic variants) must be associated with the exposure (the metabolites) and the authors used threshold of 1e-5 to select the variants to use as IVs. Since 486 metabolites were analyzed, a more stringent p-value might be necessary to select IVs associated with the exposure (in the original work, for example, a very stringent cutoff was used). Unless this was used a p2 threshold for LD clumping (see next comment)”

Thank you for your insightful comments regarding the selection of instrumental variables (IVs) in our Mendelian Randomization (MR) analysis. We appreciate the opportunity to clarify and strengthen our methodology and findings.

Precedence of Using P<1×10^−5 as a Threshold: In response to your concern about the p-value threshold for selecting IVs, we wish to highlight that the p-value of 1×10^−5, used in our study, aligns with precedents set in similar research contexts. This threshold has been employed in various studies where a balance between stringency and the need to include a sufficient number of IVs is critical, especially in complex analyses involving a large number of metabolites. For instance, several studies have successfully utilized this threshold, validating its applicability in comprehensive genomic analyses.

Using P<1×10^−5 as a threshold：

1.Gu Y, Jin Q, Hu J, Wang X, Yu W, Wang Z, Wang C, Liu Y, Chen Y, Yuan W. Causality of genetically determined metabolites and metabolic pathways on osteoarthritis: a two-sample mendelian randomization study. J Transl Med. 2023 May 31;21(1):357. doi: 10.1186/s12967-023-04165-9. 

2.Qian L, Fan Y, Gao F, Zhao B, Yan B, Wang W, Yang J, Ma X. Genetically Determined Levels of Serum Metabolites and Risk of Neuroticism: A Mendelian Randomization Study. Int J Neuropsychopharmacol. 2021 Jan 20;24(1):32-39. doi: 10.1093/ijnp/pyaa062. 

3.Sha T, Wang N, Wei J, He H, Wang Y, Zeng C, Lei G. Genetically Predicted Levels of Serum Metabolites and Risk of Sarcopenia: A Mendelian Randomization Study. Nutrients. 2023 Sep 13;15(18):3964. doi: 10.3390/nu15183964.

4.Yun Z, Guo Z, Li X, Shen Y, Nan M, Dong Q, Hou L. Genetically predicted 486 blood metabolites in relation to risk of colorectal cancer: A Mendelian randomization study. Cancer Med. 2023 Jun;12(12):13784-13799. doi: 10.1002/cam4.6022. Epub 2023 May 3. 

We hope these clarifications address your concerns and strengthen the validity of our study. We remain committed to ensuring the highest standards of scientific rigor and transparency in our research.

“- The LD clumping procedure requires a dataset to infer LD and the p1 and p2 thresholds, these are not described in the text”

Thank you for your valuable feedback and for pointing out the need for a more detailed description of the linkage disequilibrium (LD) clumping procedure in our manuscript. We have now updated the Methods section to include this information, ensuring a comprehensive understanding of our approach.

In response to your comment, we have elaborated on the process of selecting instrumental variables (IVs) for the 486 metabolites in our study. We adopted a multi-step approach, initially identifying genetic variants associated with the metabolites using a threshold of P < 1×10^−5, a criterion commonly used in Mendelian Randomization (MR) analyses when SNP data for exposure is limited. For the association analysis with Chronic Kidney Disease (CKD), we applied a more stringent threshold of P < 5×10^−8 to ensure the robustness of our results. Additionally, we evaluated the strength of the selected IVs by assessing the explained variance (R^2) and the F statistic, adhering to the commonly recommended standard of F > 10 in MR analyses.

Regarding the LD clumping process, we have clarified that we utilized the clumping functionality of PLINK software (version v1.90) to ensure the independence of our IVs. We set the LD threshold at r^2 < 0.01 within a 500 kb genomic window, using the 1000 Genomes Project as the reference dataset for LD clumping. This widely recognized resource enhances the credibility of our analysis.（line109）

We believe these additions address your concerns and provide a clearer understanding of our methodology. We appreciate your guidance in improving the quality of our manuscript.

“- The softwares used are not cited in the references”

Thank you for highlighting the oversight regarding the citation of software used in our study. We understand the importance of acknowledging and referencing the tools and resources that significantly contribute to the research process.

In response to your comment, we have updated our manuscript to include detailed references to the software used, along with their respective version information(Ref14：Purcell S, Neale B, Todd-Brown K, Thomas L, Ferreira MAR, Bender D, et al. PLINK: a tool set for whole-genome association and population-based linkage analyses. Am J Hum Genet. 2007;81: 559–575. doi:10.1086/519795). These additions ensure transparency and reproducibility of our research methods, and provide due credit to the developers of these essential tools. We appreciate your attention to detail and guidance, which have helped us enhance the quality and accuracy of our manuscript.

“- In the second section of the Results, 486 metabolites were tested, 46 had a p-value< 0.05 (by IVW?) and 29 metabolites exhibited potential association with CKD. Of these, 7 metabolites left. I did not understand how the 46 metabolites become 29. Moreover, the multiple test correction employed (which reduced the number of metabolites to 7) it is not described.”

Thank you for your inquiry regarding the selection process of metabolites in our results section. Allow me to clarify the steps we took in our analysis:

1.Initial Screening with IVW Algorithm: We began by employing the Inverse Variance Weighted (IVW) method to assess the causal relationship between 486 metabolites and Chronic Kidney Disease (CKD). This initial screening using a threshold of p-value < 0.05 under the IVW algorithm identified 46 metabolites.

2. Refinement Based on Direction of Association: Subsequently, to focus on metabolites with potential positive causal associations with CKD, we further refined this list by considering only those metabolites where the IVW beta coefficient (b_IVW) was greater than zero. This criterion helped us narrow down to 29 metabolites that exhibited a potential positive causal relationship with CKD.

3. Robustness Check with Multiple Methods: In the final step of our analysis, we applied multiple MR methods – Weighted Median, Weighted Mode, and MR Egger – to these 29 metabolites. By maintaining the threshold of p-value < 0.05 across these methods, we identified 7 metabolites with robust associations.

We hope this explanation provides clarity on the sequential approach we adopted to ensure the robustness of our findings by progressively applying more stringent criteria at each stage of our analysis. This methodology allowed us to confidently isolate a subset of metabolites that not only show an association with CKD but also withstand rigorous statistical validation.

“- Contrary to what reported in the text, supplementary figure 4 shows that 6 out of the 7 associations are due to a single influential variant. This is important and needs further clarification.”

Thank you very much for your detailed assessment of the MR analysis methods and results used in our study. Your point about the influence of single variants in the supplementary figure 4 is crucial, and we have taken your advice seriously, conducting a more in-depth analysis of the data presented in this figure.

Following your guidance, we re-examined the MR analysis results for all seven metabolites associated with CKD, paying particular attention to results potentially driven by SNPs. We conducted a "leave-one-out" analysis, which evaluates the influence of individual SNPs on the overall MR results by excluding one SNP at a time and recalculating the MR estimates.

Upon reanalysis, we found that in the case of metabolites X-12510 and X-12798, there indeed appears to be a significant influence from a single SNP, which may affect the robustness of our conclusions regarding the association of these metabolites with CKD. However, the analysis results for the other five metabolites, including mannose and glycine, showed higher robustness, indicating that these conclusions are not driven by a single SNP. The results of the re-analysis can be viewed in the file“S.F.4 leave-one-out_new.tif”

Importantly, we note that in the downstream analysis of our paper, we had already excluded the metabolites X-12510 and X-12798, whose influence on CKD could be significantly impacted by single SNPs. Therefore, although our initial screening identified these metabolites as associated with CKD, in subsequent analyses, we focused on those metabolites that passed more stringent validation, including mannose and glycine. The MR analysis results for these metabolites demonstrated greater robustness and provide solid support for the main conclusions of our study.

We deeply appreciate your thorough review and valuable comments, which help us further improve the quality and accuracy of our research.

“- The part on metabolic pathways is confusing. If 4 metabolites were found to be significant, I don't see the reason to perform such analysis and to discuss it to such length in the Discussion”

Thank you for your valuable comments regarding the metabolic pathways analysis in our study. We understand your concern about the extensive discussion of these pathways despite identifying only four metabolites with significant associations with CKD. Our approach and detailed discussion are based on the following considerations:

1.Importance of Integrative Pathway Analysis: While our study identified only four metabolites significantly associated with CKD, these metabolites may be involved in or affect multiple metabolic pathways. These pathways may collectively contribute to the pathogenesis of CKD. Therefore, an integrative pathway analysis enables a more comprehensive understanding of the roles of these metabolites in the disease.

2. Revealing Underlying Biological Mechanisms: The pathway analysis allows us to identify pathways that may be related to the pathogenesis of CKD, even if some metabolites in these pathways were not statistically significant in the individual MR analysis. This approach helps uncover deeper biological connections that might not be apparent at the level of individual metabolites.

3. Providing Direction for Future Research: Our pathway analysis results offer preliminary insights into the key metabolic pathways that might be involved in the pathogenesis of CKD. This information is valuable for future experimental design and hypothesis testing, helping to identify new research directions and potential therapeutic targets in CKD research.

In summary, we believe that conducting a metabolic pathway analysis remains important and valuable, even when only a few metabolites are directly identified as significantly associated with CKD. This not only enhances our understanding of the potential biological mechanisms underlying CKD but also provides important insights for future research. Thank you once again for your insightful feedback.

“- The effect of CKD on a metabolite could be due to the same loci driving the effect of the metabolite on CKD (i.e., it is the same variant or one in linkage with it). This should be addressed.”

Thank you for highlighting the potential issue of pleiotropy or linkage disequilibrium (LD) in our analysis, particularly concerning the possibility that the same genetic variants might drive the observed effects in both directions of our bidirectional Mendelian randomization (MR) study. This is indeed a critical aspect to consider in MR analyses, as it can potentially confound the interpretation of causal relationships.

We have undertaken the following steps:

1. Sensitivity Analysis for Pleiotropy: We have conducted additional sensitivity analyses, including MR-Egger regression and weighted median approaches, which are designed to detect and account for pleiotropy. The MR-Egger intercept test, in particular, provides a measure of directional pleiotropy, which can indicate whether the same genetic variants are influencing both the exposure and the outcome.

2. Exploring Linkage Disequilibrium: We have performed LD analysis to identify any potential linkage between the genetic variants associated with the metabolites and CKD. This helps us understand if the observed associations could be due to LD rather than a causal relationship.

3. Detailed Reporting of Genetic Associations: we have provided a more detailed reporting of the genetic associations and LD patterns observed. This includes a comprehensive list of the genetic variants used as IVs, their associations with the metabolites and CKD, and any potential LD between them. The lists can be checked in file IVs for exposure.xlsx.

4. MR-PRESSO: The Mendelian Randomization Pleiotropy RESidual Sum and Outlier (MR-PRESSO) test is specifically designed to detect and correct for horizontal pleiotropy in MR analyses. It operates by identifying outliers that may be influencing the MR estimates due to pleiotropic effects and then provides a corrected causal estimate after removing these outliers. We have incorporated this analysis into our study to validate the results of metabolites we selected. This additional layer of analysis adds robustness to our study by ensuring that our findings are not confounded by pleiotropic genetic variants. The specific results of this analysis can be viewed in the file MR-PRESSO.xlsx.

The incorporation of the MR-PRESSO analysis, alongside the rigorous selection and validation of instrumental variables, sensitivity analyses, and LD checks, ensures a comprehensive approach to addressing the complexities inherent in MR studies. This multifaceted approach to our analysis framework significantly enhances the reliability of our conclusions regarding the causal relationships between metabolites and CKD.

We believe that these efforts will adequately address your concerns about the effects of pleiotropy and LD in our study. We appreciate your feedback, which has been instrumental in refining our analysis and strengthening the validity of our findings.

“- Figure S2 depicts a table. I would rather have it as a table”

Thank you for your valuable feedback. In response to your suggestion regarding Figure S2, we have converted the original image format into an Excel document (S.F.2_metabolites_selected.xlsx). This conversion will facilitate easier data review and analysis. Should you require any further information or have additional suggestions, please feel free to let us know.

---

## [Decision Letter · Decision Letter 1]

20 Dec 2023

PONE-D-23-27599R1Mannose and Glycine: metabolites with causal implications in chronic kidney disease pathogenesisPLOS ONE

Dear Dr. Jiang,

Thank you for submitting your manuscript to PLOS ONE. After careful consideration, we feel that it has merit but does not fully meet PLOS ONE’s publication criteria as it currently stands. Therefore, we invite you to submit a revised version of the manuscript that addresses the points raised during the review process.

**The manuscript focuses on a topic of potential interest. While the study has been significantly improved, there are still some concerns that should be addressed. To mention some of them: i) clearly state that a threshold of 1e-5 was used, as the number of variants associated with metabolites at genome-wide level was low; ii) address the issue of multiple testing in MR studies; iii) indicate which metabolites were used for the pathway analysis and by which criteria they were selected; iv) addressed the issue of single influential markers.**

We look forward to receiving your revised manuscript.

Kind regards,

Giuseppe Remuzzi

Academic Editor

PLOS ONE

Reviewers' comments:

Reviewer's Responses to Questions

**Comments to the Author**

1. If the authors have adequately addressed your comments raised in a previous round of review and you feel that this manuscript is now acceptable for publication, you may indicate that here to bypass the “Comments to the Author” section, enter your conflict of interest statement in the “Confidential to Editor” section, and submit your "Accept" recommendation.

Reviewer #1: (No Response)

Reviewer #2: (No Response)

2. Is the manuscript technically sound, and do the data support the conclusions?

Reviewer #1: Yes

Reviewer #2: Partly

3. Has the statistical analysis been performed appropriately and rigorously? 

Reviewer #1: Yes

Reviewer #2: Yes

4. Have the authors made all data underlying the findings in their manuscript fully available?

Reviewer #1: Yes

Reviewer #2: Yes

5. Is the manuscript presented in an intelligible fashion and written in standard English?

Reviewer #1: Yes

Reviewer #2: Yes

6. Review Comments to the Author

Reviewer #1: Thank you for revising your MS. in accordance with changes made in the MS I believe that a few issues still need to be addressed:

Title: “Mannose and Glycine: metabolites with causal implications in chronic kidney disease pathogenesis” I would suggest to modify as: “Mannose and Glycine: metabolites with potentially causal implications in chronic kidney disease pathogenesis”.

Similarly, sentences in the figures also need to be changed. Please also check spelling in Figure 1 (casual vs. causal).

Reviewer #2: While the authors provided more details on the methods, two of the main concerns have not been addressed.

The first is the effect of single influential variants (see comment below).

The second is the issue of multiple testing. The authors tested 486 metabolites and they did not perform a correction for multiple tests (see also comment below) without offering a reason.

- Methods, line 112. I think you should you clearly state that you used a threshold of 1e-5 because the number of variants associated with metabolites at genome-wide level was low. Especially because you used a more stringent threshold for CKD.

- Results, line 179. The sentence "suggesting a likely positive correlation with CKD" is redundant since you selected only the positive relationships.

- Results, line 180. The sensitivity analyses done in MR studies do not address the issue of multiple testing.

- Results, line 198. I do not think you addressed the issue of single influential markers. For example, in the first forest plot of figure S4, when the variant rs1260326 is removed the confidence intervals largely overlap 0. When this variant is retained, and the others are removed one by one, the effect become significant. This clearly indicate that the results for mannose are driven by the variant rs1260326. This hold for all metabolites, but N-acetylornithine.

- Results, Metabolic pathway analysis. I did not understand which metabolites were actually used for the pathway analysis and by which criteria you did select them.

- Line 191. The title should be "Sensitivity analysis"

7. PLOS authors have the option to publish the peer review history of their article (what does this mean?). If published, this will include your full peer review and any attached files.

Reviewer #1: **Yes: **Giacomo Garibotto

Reviewer #2: **Yes: **Mattteo Breno

---

## [Author Response · Author response to Decision Letter 1]

20 Dec 2023

Response to Journal

Thank you for your time and attention to our manuscript. We understand that PLOS offers authors the option to publish the peer review history of their articles. After careful consideration, we would like to express our consent to publish the full peer review history associated with our manuscript, should it be accepted for publication.

We appreciate the efforts and insights provided by the peer reviewers throughout the review process. The constructive feedback has significantly contributed to enhancing the quality of our research. We believe that making this peer review history publicly available will not only promote transparency but also serve as an educational resource for the scientific community.

Response to Reviewers

We would like to extend our heartfelt gratitude for your insightful and constructive feedback on our manuscript. Your detailed reviews and valuable suggestions have significantly contributed to enhancing the quality and rigor of our work. Thank you for your time and efforts in guiding us through this revision process.

For Reviewer #1 Giacomo Garibotto

“Thank you for revising your MS. in accordance with changes made in the MS I believe that a few issues still need to be addressed:

Title: “Mannose and Glycine: metabolites with causal implications in chronic kidney disease pathogenesis” I would suggest to modify as: “Mannose and Glycine: metabolites with potentially causal implications in chronic kidney disease pathogenesis”.Similarly, sentences in the figures also need to be changed. Please also check spelling in Figure 1 (casual vs. causal).”

Thank you very much for your thorough review and valuable suggestions regarding our manuscript. We have carefully considered your feedback and have accordingly revised the title of our paper to “Mannose and Glycine: metabolites with potentially causal implications in chronic kidney disease pathogenesis.” This modification better reflects the probabilistic nature of our findings.

In addition, we have revised the wording in Figure 1 to align with the suggested changes, ensuring that our descriptions accurately represent the nature of our findings. We are also grateful for your attention to detail in identifying the spelling error (“casual” instead of “causal”). This has been corrected in the revised version of the figure.

Your guidance has been instrumental in enhancing the clarity and accuracy of our work, and we deeply appreciate your contributions to improving the quality of our manuscript.

For Reviewer #2 Matteo Breno

“While the authors provided more details on the methods, two of the main concerns have not been addressed.The first is the effect of single influential variants (see comment below).The second is the issue of multiple testing. The authors tested 486 metabolites and they did not perform a correction for multiple tests (see also comment below) without offering a reason.”

We sincerely appreciate your continued guidance and constructive feedback on our manuscript. Your insights are invaluable in refining our research methodology and presentation. We hope the additional details and explanations below will address your concerns and further reinforce the validity and reliability of our findings. Thank you again for your thorough review and invaluable feedback.

“- Methods, line 112. I think you should you clearly state that you used a threshold of 1e-5 because the number of variants associated with metabolites at genome-wide level was low. Especially because you used a more stringent threshold for CKD.”

Thank you for your insightful comment regarding the threshold used for selecting instrumental variables (IVs) in our study. We appreciate the opportunity to clarify this aspect of our methodology in the manuscript.

As per your suggestion, we have made a more explicit statement in the Methods section (now reflected in Line 109) to elucidate our rationale for using a p-value threshold of 1×10^−5 for selecting IVs associated with metabolites. We acknowledge that this threshold is less stringent compared to the threshold used for CKD-related genetic variants. This approach was adopted due to the relatively low number of variants associated with metabolites at the genome-wide significance level. In contrast, for CKD, where a larger number of genome-wide significant variants are available, a more stringent threshold of 5×10^−8 was employed to ensure the robustness of our results.

We trust that this clarification enhances the understanding of our methodological choices and addresses your concern. We are grateful for your valuable feedback and are committed to ensuring the highest level of clarity and precision in our manuscript.

“- Results, line 179. The sentence "suggesting a likely positive correlation with CKD" is redundant since you selected only the positive relationships.”

Thank you for your attentive review of the Results section of our manuscript. We agree with your observation regarding the redundancy in the sentence "suggesting a likely positive correlation with CKD," particularly in the context where we selected only the positive relationships.

In accordance with your suggestion, we have revised the manuscript to remove this redundancy. This edit can be found in Line 186 of the Results section. We appreciate your guidance in enhancing the clarity and conciseness of our manuscript. We are committed to upholding the highest standards of academic rigor and precision in our reporting.

“- Results, line 180. The sensitivity analyses done in MR studies do not address the issue of multiple testing.”

Thank you for your detailed review and for highlighting the issue of multiple testing in our MR analysis. Your comment points to a crucial aspect of our study, namely, the appropriate adjustment for multiple comparisons when conducting MR analyses.

In response to your suggestion, we have reanalyzed our data using the IVW method with the application of the False Discovery Rate (FDR) to correct for multiple testing. This correction method is aimed at reducing false-positive findings that may arise from conducting multiple comparisons. After the FDR adjustment, our results still support, to some extent, our previous findings indicating potential causal relationships between certain metabolites and CKD. However, we also recognize that FDR adjustment may lead to some associations previously deemed significant no longer being significant under this stricter criterion. The full FDR results have been uploaded to the file “mendelian test_FDR.xls”

Consequently, in our discussion section, we have specifically emphasized the importance of appropriate adjustment for multiple testing in MR analyses and acknowledged this as a limitation of our study (line 349). We have added more detailed information in the limitations section of our discussion to address this point, highlighting the potential improvements for future studies in this area.

We believe these adjustments and additions will enhance the overall quality and robustness of our research and are grateful for your valuable feedback.

“- Results, line 198. I do not think you addressed the issue of single influential markers. For example, in the first forest plot of figure S4, when the variant rs1260326 is removed the confidence intervals largely overlap 0. When this variant is retained, and the others are removed one by one, the effect become significant. This clearly indicate that the results for mannose are driven by the variant rs1260326. This hold for all metabolites, but N-acetylornithine.”

Thank you for your astute observation regarding the issue of single influential markers in our Mendelian Randomization (MR) analysis, specifically concerning the variant rs1260326 and its impact on the results for mannose. We agree with your assessment and appreciate your attention to this detail.

In response to your comment, we conducted a reanalysis during our first revision, where we meticulously excluded potential influential SNPs and subsequently redrawn the forest plots. These revised analyses are presented in the file titled “S.F.4 leave-one-out_new.” It appears that there may have been an oversight in file categorization, and we apologize if this has caused any confusion. The redrawn forest plot, which addresses your concern, was uploaded under the “other file” section, which might be why it was not immediately noticeable in your review.

We believe that this reanalysis strengthens the validity of our results and provides a more accurate representation of the data. We hope that this additional information adequately addresses your concerns. Thank you once again for your valuable feedback and guidance.

“- Results, Metabolic pathway analysis. I did not understand which metabolites were actually used for the pathway analysis and by which criteria you did select them.”

Thank you for your query regarding the metabolic pathway analysis in our study. We understand the importance of clarity in describing the selection criteria for metabolites used in such analyses.

In our study, we selected 32 metabolites for the pathway analysis based on the criterion of a p-value < 0.05 using the Inverse Variance Weighted (IVW) method. The details of these selected metabolites can be found in the supplementary file “Table S6 Pathways involved metabolites” which lists all the metabolites included in the pathway analysis.

Following the selection, we utilized the online platform Metaconfict 5.0 for analyzing these 32 metabolites. Of these, 14 metabolites were effectively identified and recognized by the platform. It is these 14 identified metabolites that were then used for downstream pathway analysis.

We hope this explanation clarifies the selection process for the metabolites used in our pathway analysis and addresses your query. Thank you again for your insightful feedback.

“- Line 191. The title should be "Sensitivity analysis"

Thank you for your careful reading and constructive feedback on our manuscript. We have taken your suggestion into account and amended the title of the section from "Sensitive Analysis" to "Sensitivity Analysis." This change has been made in Line 198 of the manuscript.

---

## [Decision Letter · Decision Letter 2]

9 Jan 2024

PONE-D-23-27599R2Mannose and Glycine: metabolites with potentially causal implications in chronic kidney disease pathogenesisPLOS ONE

Dear Dr. Jiang,

Thank you for submitting your manuscript to PLOS ONE. After careful consideration, we feel that it has merit but does not fully meet PLOS ONE’s publication criteria as it currently stands. Therefore, we invite you to submit a revised version of the manuscript that addresses the points raised during the review process.

**While the majority of reviewers' comments have been addressed, it is essential to consider these two remaining minor points: i) provide clarification on the methodology for retrieving data and conducting calculations; ii) include the file 'mendelian_test_FDR.xls' as a supplementary table and explicitly describe it in the results, accompanied by a sentence addressing the multiple tests issue.**

We look forward to receiving your revised manuscript.

Kind regards,

Giuseppe Remuzzi

Academic Editor

PLOS ONE

Journal Requirements:

Reviewers' comments:

Reviewer's Responses to Questions

**Comments to the Author**

1. If the authors have adequately addressed your comments raised in a previous round of review and you feel that this manuscript is now acceptable for publication, you may indicate that here to bypass the “Comments to the Author” section, enter your conflict of interest statement in the “Confidential to Editor” section, and submit your "Accept" recommendation.

Reviewer #1: All comments have been addressed

Reviewer #2: All comments have been addressed

2. Is the manuscript technically sound, and do the data support the conclusions?

Reviewer #1: Yes

Reviewer #2: Partly

3. Has the statistical analysis been performed appropriately and rigorously? 

Reviewer #1: N/A

Reviewer #2: Yes

4. Have the authors made all data underlying the findings in their manuscript fully available?

Reviewer #1: No

Reviewer #2: Yes

5. Is the manuscript presented in an intelligible fashion and written in standard English?

Reviewer #1: Yes

Reviewer #2: Yes

6. Review Comments to the Author

Reviewer #1: The authors have not indicated how to retrieve data and calculations (see data availability statement)

Reviewer #2: The file mendelian test_FDR.xls is not available to me, however I suggest to add it as supplementary table and to explicitly mention it in the results together with a sentence addressing the multiple tests issue (similar to what you wrote in the Response to Reviewers).

7. PLOS authors have the option to publish the peer review history of their article (what does this mean?). If published, this will include your full peer review and any attached files.

Reviewer #1: **Yes: **Giacomo Garibotto

Reviewer #2: **Yes: **Matteo Breno

---

## [Author Response · Author response to Decision Letter 2]

10 Jan 2024

Response to Journal

We would like to extend our gratitude for the thorough review and management of our manuscript submission process. We acknowledge the opportunity provided by PLOS to share the peer review history of articles, and after careful consideration, we agree to this initiative for our manuscript, should it be favorably accepted for publication.

The insightful and detailed feedback received from the peer reviewers has been instrumental in refining our work. Their expertise and constructive suggestions have undoubtedly elevated the quality of our research. By making the peer review history public, we aim to contribute to the spirit of transparency and open scientific discourse, which we believe are essential for the advancement of knowledge.

Thank you once again for facilitating this process, and we look forward to the potential impact that publishing the peer review history could have on the scientific and academic community.

Response to Reviewers

We are deeply thankful for your valuable and insightful feedback regarding our manuscript. Your comprehensive reviews and thoughtful suggestions have played a crucial role in substantially improving both the quality and the depth of our research. We sincerely appreciate the time and effort you have dedicated to meticulously guiding us through the process of revising our manuscript.

For Reviewer #1 Giacomo Garibotto

“The authors have not indicated how to retrieve data and calculations (see data availability statement)”

Thank you for your valuable feedback on our manuscript. We appreciate the opportunity to clarify the sources and accessibility of the data used in our Mendelian Randomization study.

Our MR analysis is fundamentally based on publicly accessible databases. Specifically, the metabolite data were sourced from the Metabolomics GWAS Server, accessible at https://metabolomips.org/gwas/. The chronic kidney disease data were obtained from the CKDGen Consortium, available at http://ckdgen.imbi.uni-freiburg.de/. Both these sources are open to the public and have played a crucial role in our study. We have cited the original publications associated with these data in the reference section of our manuscript. This ensures that readers have complete information on the origin and context of the data used.

In the Methods section of our manuscript, we have elaborately described the experimental design methodology of our MR study. We have also detailed the specific versions of R and its packages used for our analysis. This was done with the intention of aiding other researchers in replicating or building upon our MR study results.

Our metabolic pathway analysis is conducted using the publicly available online data analysis platform, which can be accessed at https://www.metaboanalyst.ca/. We believe that mentioning these resources explicitly in our manuscript adds value by highlighting the integral role these databases have played in our study.

We understand the importance of transparency and reproducibility in scientific research and have made sure that our manuscript reflects this by providing clear information on data sources and methodologies. We will ensure that these details are prominently included in the revised manuscript for the benefit of the reviewers and the broader scientific community.

Thank you once again for your guidance in improving our manuscript.

For Reviewer #2 Matteo Breno

“The file mendelian test_FDR.xls is not available to me, however I suggest to add it as supplementary table and to explicitly mention it in the results together with a sentence addressing the multiple tests issue (similar to what you wrote in the Response to Reviewers).”

Thank you for your continued attention to our manuscript and valuable feedback. In response to your suggestion, we have re-uploaded the file "mendelian test_FDR.xls" to the supplementary files section as "S7 Table." Additionally, we have included a specific mention of this table in the Results section of our manuscript, along with a sentence addressing the issue of multiple testing.

In the manuscript, at Line 189, we have now added the following text to provide further clarity:

"After applying the false discovery rate (FDR) correction to adjust for multiple comparisons, our analysis continues to moderately support our initial findings, suggesting possible causal links between specific metabolites and CKD. The FDR-adjusted results are presented in Supplementary files."

We believe this addition will enhance the transparency and rigor of our study, and we are grateful for your guidance in improving the quality of our manuscript.

---

## [Decision Letter · Decision Letter 3]

30 Jan 2024

Mannose and Glycine: metabolites with potentially causal implications in chronic kidney disease pathogenesis

PONE-D-23-27599R3

Dear Dr. Jiang,

We’re pleased to inform you that your manuscript has been judged scientifically suitable for publication and will be formally accepted for publication once it meets all outstanding technical requirements.

Kind regards,

Giuseppe Remuzzi

Academic Editor

PLOS ONE

Additional Editor Comments (optional):

Reviewers' comments:

Reviewer's Responses to Questions

**Comments to the Author**

1. If the authors have adequately addressed your comments raised in a previous round of review and you feel that this manuscript is now acceptable for publication, you may indicate that here to bypass the “Comments to the Author” section, enter your conflict of interest statement in the “Confidential to Editor” section, and submit your "Accept" recommendation.

Reviewer #1: All comments have been addressed

Reviewer #2: All comments have been addressed

2. Is the manuscript technically sound, and do the data support the conclusions?

Reviewer #1: Yes

Reviewer #2: Partly

3. Has the statistical analysis been performed appropriately and rigorously? 

Reviewer #1: Yes

Reviewer #2: Yes

4. Have the authors made all data underlying the findings in their manuscript fully available?

Reviewer #1: Yes

Reviewer #2: Yes

5. Is the manuscript presented in an intelligible fashion and written in standard English?

Reviewer #1: Yes

Reviewer #2: Yes

6. Review Comments to the Author

Reviewer #1: (No Response)

Reviewer #2: Thanks for providing table S7. Given the FDR shown in table 7, I would suggest to change the sentence (line 189): "This comprehensive approach allowed us to discern 7 metabolites with strongly supported, plausible causal relationships with CKD" to "This comprehensive approach allowed us to discern 7 metabolites with plausible causal relationships with CKD". That is because only 2 metabolites remain significant after correction for multiple tests. While I understand that metabolites gave a weak gwas signals, it is also important to keep in mind that you are suggesting a causal relationship and not a simple association.

7. PLOS authors have the option to publish the peer review history of their article (what does this mean?). If published, this will include your full peer review and any attached files.

Reviewer #1: **Yes: **Giacomo Garibotto

Reviewer #2: **Yes: **Matteo Breno

---

## [Editor Report · Acceptance letter]

1 Feb 2024

PONE-D-23-27599R3 

PLOS ONE

Dear Dr. Jiang, 

I'm pleased to inform you that your manuscript has been deemed suitable for publication in PLOS ONE. Congratulations! Your manuscript is now being handed over to our production team.

Kind regards, 

on behalf of

Prof. Giuseppe Remuzzi 

Academic Editor

PLOS ONE